# A Novel Extracellular Catalase Produced by the Antarctic Filamentous Fungus *Penicillium Rubens* III11-2

**Zdravka Koleva** [1] , **Radoslav Abrashev** [2], **Maria Angelova** [2], **Galina Stoyancheva** [3], **Boryana Spassova** [2], **Lyudmila Yovchevska** [2], **Vladislava Dishliyska** [2], **Jeny Miteva-Staleva** [2] and **Ekaterina Krumova** [2,*]

[1] Department of Natural Sciences, New Bulgarian University, 1618 Sofia, Bulgaria; zdrawka_kolewa@abv.bg
[2] Department of Mycology, The Stephan Angeloff Institute of Microbiology, Bulgarian Academy of Sciences, Acad. G. Bonchev Str. Bl. 26, 1113 Sofia, Bulgaria; rabrashev@microbio.bas.bg (R.A.); mariange@microbio.bas.bg (M.A.); bkasovska@abv.bg (B.S.); lyovchevska@abv.bg (L.Y.); vladydacheva@yahoo.com (V.D.); j_m@abv.bg (J.M.-S.)
[3] Department of General Microbiology, The Stephan Angeloff Institute of Microbiology, Bulgarian Academy of Sciences, Acad. G. Bonchev Str. Bl. 26, 1113 Sofia, Bulgaria; galinadinkova@abv.bg
* Correspondence: e_krumova@microbio.bas.bg

**Abstract:** Catalase (CAT) is an enzyme involved in the first line of cellular antioxidant defense. It plays a key role in the protection of a wide range of Antarctic organisms against cold stress. Extracellular catalase is very rare and data on it are extremely scarce. The aim of the present study was to select an efficient producer of extracellular catalase from amongst Antarctic filamentous fungi. Sixty-two Antarctic filamentous fungal strains were investigated for their potential ability to synthesize intracellular and extracellular CAT. The Antarctic strain *Penicillium rubens* III11-2 was selected as the best producer of extracellular catalase. New information on the involvement of the extracellular antioxidant enzymes superoxide dismutase and CAT in the response of filamentous fungi against low-temperature stress was obtained. An efficient scheme for the purification of CAT from culture fluid was developed. An enzyme preparation with high specific activity (513 U/mg protein) was obtained with a yield of 19.97% and a purification rate of 98.4-fold. The purified enzyme exhibited maximal enzymatic activity in the temperature range of 5–40 °C and temperature stability between 0 and 30 °C, therefore being characterized as temperature sensitive. To our knowledge, this is the first purified extracellular cold active catalase preparation from Antarctic filamentous fungi.

**Keywords:** Antarctic; filamentous fungi; extracellular catalase; *Penicillium rubens*



## 1. Introduction

Habitats with extremely low temperatures are a poorly explored niche for producers of biologically active substances. One such habitat where filamentous fungi of various temperature classes have been isolated is Antarctica [1]. Filamentous fungi inhabiting that continent are exposed to a wide range of extreme environmental conditions, including drought, high salinity, solar radiation and low temperatures. They can also be utilized as sources of novel and useful primary and secondary metabolites such as enzymes.

Cold-adapted microorganisms have evolved different strategies to adapt to low temperatures [1–3]. The synthesis of temperature-sensitive (cold-active, CA) enzymes [4–6], the modification of lipid composition to preserve cell membrane fluidity [7], the existence of RNA chaperones that prevent the formation of undesirable RNA secondary structures [8,9], the synthesis of antifreeze proteins [10–12], and other processes are all part of adaptation. Furthermore, since low temperatures lead to the production of reactive oxygen species (ROS) and cause oxidative stress, a higher level of antioxidant defense also contributes to increased tolerance to low-temperature stress [10,11]. The first line of antioxidant defense includes the enzymes superoxide dismutase (SOD) and CAT, which play an essential role in inducing resistance to various stressors, including temperature [12,13].

Catalases produced by fungi are heme-containing enzymes, namely typical monofunctional CATs and bifunctional catalase-peroxidases, and also Mn-containing ones. Several strains, such as *Aspegillus nidulans* [14], *A. phoenicis* [15], *Penicillium chrysogenum* [16], *P. marneffei* [17], *Alternaria alternata* [18], etc., have been reported to be CAT producers. MTCC 6324, a monofunctional enzyme with a molecular mass of 368 kDa, was isolated from *A. terreus* [19]. A purified enzyme was also obtained from *Neurospora crassa* [20], *Agaricus bisporus* [21], the opportunistic fungus *Scedosporium boydii* [22], and the mycorrhizal fungus *Terfezia claveryi* TcCAT-1 [23]. Fiedurek and Gromada (2000) developed a method to increase the CAT level by increasing the dissolved oxygen (DO) concentration in the culture filtrate [24].

Producers isolated from habitats with extremely low temperatures can synthesize CA enzymes, including CAT [25,26]. CA enzymes from psychrophilic or psychrotolerant bacteria or fungi exhibit optimal activity at temperatures of 20–45 °C and maintain thermal stability at much lower temperatures. These enzymes are more thermolabile compared to their mesophilic homologs and possess a more flexible structure, resulting in higher catalytic efficiency, i.e., an increase in the number of catalytic cycles to achieve thermal compensation. The noted properties provide greater opportunities for structural changes during the process [27]. The main advantages of CA enzymes are high specific activity, lower stability, and unusually high substrate specificity.

However, relatively little is known about CA catalase enzymes from filamentous fungi. Research on CA enzyme synthesis is mostly focused on bacterial species. A facultatively psychrophilic strain of *Vibrio rumoiensis* [28], a psychrophilic marine strain of *V. salmonicida* LFI1238 [29], and a mesophilic strain of *Serratia marcescens* SYBC-01 [30] have all been reported to produce CA CAT. In cell extracts from Antarctic bacteria, CA CAT has been identified and extracted [31]. The structure and catalytic efficiency of CA CAT from *V. salmonicida* were reported by Riise et al. (2007), who also explained the great flexibility and decreased thermal stability of the compound [32]. The catalytic function of psychrophilic CAT was determined by comparing the amino acid composition of CA CAT from Antarctic *Bacillus* sp. N2a with its meso-philic mesophilic counterpart from *Bacillus subtilis* TE124. There are few reports on CA catalase produced by Antarctic fungi [24]. The second published information is by Krumova et al. (2021) on CA intracellular CAT produced by an Antarctic strain of *P. griseofulvum* p 29 [33]. The authors developed a laboratory technology for the production of the enzyme and purified and characterized the resulting enzyme.

As it is known, CAT is found in the cytoplasm or peroxisomes, but extracellularly synthesized enzymes have also been reported. The role of the extracellular enzyme is to protect cells against exogenous oxidative damage. From an industrial point of view, the secretion of CAT into the culture filtrate is a more cost-effective option that allows easier and cheaper purification. Microorganisms, particularly filamentous fungal strains, are the most effective CAT producers in the extracellular environment [34–36]. *P. cyclopium* isolated from the Arctic tundra generated both enzymes, and Fiedurek et al. (2003) reported that the ratio of intracellular to extracellular catalase was 1:3 [37]. Kacem-Chaouche et al. (2005) conducted similar experiments [15]. Furthermore, little is known about the properties of extracellular CAT of Antarctic origin.

Important unresolved problems in this area concern the role of the extracellular antioxidant enzyme CAT in the cellular response to oxidative stress in the survival mechanism of microorganisms isolated from permanently cold habitats, and certain main characteristics of a purified novel microbial CA enzyme CAT.

The main objective of this paper is to present data on the synthesis of extracellular catalase by Antarctic filamentous fungi and its role in their survival under low-temperature conditions. In this study, an effective scheme for enzyme purification was developed and some characteristics of purified CA CAT were obtained.

## 2. Materials and Methods

### 2.1. Microorganisms and Identification

Filamentous fungi from the Mycological Collection of the Institute of Microbiology, BAS, were used in the present work. Fungi were isolated from Antarctic soil samples collected using sterile techniques during the Antarctic summer on Livingstone Island by Bulgarian expeditions 2006 and 2007, and from Terra Nova Bay samples collected by Austrian expedition 2006.

Molecular Methods for Identification of Isolates

Chromosomal DNA was isolated from 200 mg of mycelium cultured for 48 h for each of the isolates using the Genomic DNA Kit (Macherey-Nagel, Duren, Germany). PCR reaction was conducted with universal primers for the ITS region (ITS1: 5′-TCCGTAGGTGAACCTGCGG-3′ and ITS4: 5′-TCCTCCGCTTATTGATATGC-3′) [38] in a BioRad iCycler Thermal Cycler (Bio-Rad Laboratories, Hercules, CA, USA) using a PCR master mix (GenetBio, Daejeon, Republic of Korea). Visualization of DNA and PCR products was performed using a 1% agarose gel, TAE buffer, and GelRedTM Nucleic Acid Gel Stain (Biotium, Fremont, QC, Canada). The PCR products were purified using the Gene JET PCR Purification Kit (Thermo Fisher Scientific Inc., Paisley, UK) and were sequenced at Macrogen Inc. (Amsterdam, The Netherlands). Sequence comparison was conducted using the BLAST application at NCBI (National Center for Biotechnology Information, Bethesda, MA, USA). All sequences obtained in this study were deposited in the GenBank database under accession numbers.

### 2.2. Screening for Catalase Activity

2.2.1. Cultivation on Solid Media

The model strain was cultivated on Potato dextrose agar (PDA), Sabouraud agar (SA) and Czapec dox agar (CzDA) in order to obtain data on its morphology.

2.2.2. Submerged Cultivation

*Preparation of inoculum*

For inoculum preparation, a spore suspension of 7-day-old cultures on PDA (25 °C) was used. The inoculum was cultured in 300 mL Erlenmeyer flasks with 54 mL of the medium 4/4 and 6 mL of the spore suspension on a rotary shaker (220 rpm) at 25 °C for 24 h.

*Culture conditions*

The strains were cultured in 500 mL Erlenmeyer flasks by adding 6 mL of the inoculum to 74 mL of 4/4 medium. The flasks were placed on a rotary shaker (220 rpm) at 25 °C for 72 h.

*Media for submerged cultivation of filamentous fungi*

For submerged cultivation of filamentous fungi, 4/4 medium with the following composition was used: Glucose 48.0 g/L; Soy broth 4.5 g/L; Casein 6.5 g/L; $KH_2PO_4$ 1.55 g/L; $FeSO_4$ 0.0068 g/L; $MnSO_4$ 0.0025 g/L; $ZnSO_4$ 0.007 g/L; $MgSO_4$ 0.98 g/L; $CuSO_4$ 0.69 g/L; KCl 0.0011 g/L; pH 6.0.

*Preparation of culture liquid*

After fermentation stopped, the culture liquid and mycelium were separated. The resulting liquid culture was tested for catalase activity. The mycelium was frozen at −20 °C and used for intracellular content studies as described at Abrashev et al. [25].

### 2.3. Analytical Methods—Determination of Enzyme Activities, Sugars, Protein, Dry Weight

SOD activity (EC 1.15.1.1.) was determined by the method of Beaushap and Fridovich (1971) [39]. It was assumed that one unit of SOD activity, expressed in U/mg of protein, was the quantity of enzyme protein needed to impede 50% reduction of Nitro blue tetrazolium chloride (NBT) (A560) at pH 7.8 and 30 °C. CAT activity (EC 1.11.1.6.) was determined according to the method of Beers and Sizer (1952) and expressed in U/mg of protein [40]. Absorbance was measured at 240 nm relative to the control (1 mL of suitably diluted culture

fluid with 0.5 mL of phosphate buffer pH 7.0 added). One unit of activity was assumed to be the amount of enzyme protein required to degrade 1 mM $H_2O_2$ in 1 min at 25 °C and pH 7.0.

The Somogyi–Nelson method [41] was used to determine soluble reducing sugars. Absorbance was measured at 660 nm against the control with distilled water instead of culture liquid. The glucose concentration was determined from a standard curve.

Protein concentration was measured by the method of Lowry et al. (1951) [42]. Bovine serum albumin was used as the standard. The method is colorimetric and absorbance was measured at 760 nm against a distilled water control instead of a sample. The amount of protein was determined by comparing it to a standard curve.

Upon completion of culturing, the culture fluid was filtered using a Schott filter, and then the cells were washed repeatedly with water and dried at 105 °C to constant weight.

### 2.4. Isolation and Purification of Temperature-Sensitive Catalase

To purify extracellular catalase, the strain *Penicillium rubens* III11-2 was cultivated in 500 mL Erlenmeyer flasks with medium 4/4 at 25 °C for 72 h. The culture filtrate was separated from the biomass using a Schott filter. Subsequently, the crude enzyme was fractionated using 10 kDa centrifuge filter tubes. The cooled fraction with a molecular weight of above 10 kDa was subjected to 40% ammonium sulphate saturation. After buffering (0.01 M ammonium phosphate buffer at pH 7.5), the isolated fraction was further purified with an FPLC system (ÄKTA purifier GE Healthcare Life Sciences, Suwanee, Atlanta, GA, USA) using a Q-Sepharose column (Hi prep Q FF 16/10), pre-equilibrated with 20 mM Tris HCl buffer with pH 7.8 including 100 mM NaCl. Unbound protein was eluted with the same buffer; bound enzyme protein was eluted with potassium phosphate buffers with a concentration gradient in the range from 0.01 to 0.2 M followed by 1 M potassium chloride. At each stage of purification, active fractions were analyzed for CAT activity. The CAT-rich fractions were pooled, concentrated using centrifugation in tubes with a filter permeability of 10 kDa, and stored at −20 °C for further experiments.

Visualization of purified enzyme was made using SDS PAGE; 12% gel (Laemmli method, 1971) [43] and Native PAGE (Laemmli 1970, method for visualization Woodberry) [44].

### 2.5. Determination of Temperature Optimum and Temperature Stability of the Purified Enzyme

2.5.1. Determination of Temperature Optimum

The temperature optimum of the resulting enzyme was determined by measuring the enzyme activity at temperatures of 5, 10, 20, 30, 40, 50, 60, 70 and 80 °C.

2.5.2. Determination of Temperature Stability

The temperature stability of the resulting enzyme was determined by measuring enzyme activity after treatment with temperatures of 5, 10, 20, 30, 40, 50, 60, 70 and 80 °C for 10 and 30 min, respectively.

### 2.6. Statistical Evaluation of the Results

All results presented in this study were obtained and evaluated from experiments with at least three replicates using three parallel runs. Statistical evaluation of results was performed using Student's *t* test (*t* test) for MIE (mean interval estimate), analysis of variance (ANOVA), and Dunnet's post test, with a significance level of 0.05.

## 3. Results

### 3.1. Investigation on the Potential of Antarctic Filamentous Fungi for the Catalase Synthesis

The potential of 62 Antarctic filamentous fungal strains for their ability to synthesize intracellular and extracellular CAT after 72 h of cultivation was investigated (Table 1).

**Table 1.** Screening for catalase producers.

| N | Strain | Extracell. CAT (U/mg) | Intracell. CAT (U/mg) | N | Strain | Extracell. CAT (U/mg) | Intracell. CAT (U/mg) | N | Strain | Extracell. CAT (U/mg) | Intracell. CAT (U/mg) |
|---|---|---|---|---|---|---|---|---|---|---|---|
| 1. | P27 | 2.5 | 22.9 | 22. | I 9 | 3.09 | 5.72 | 43. | I 13 | 4.25 | 8.7 |
| 2. | P22 | 3.6 | 16.5 | 23. | III $2_1$ | 3.20 | 8.16 | 44. | I $1_{12}$ | 2.27 | 4.7 |
| 3. | III $6_3$ | 2.3 | 14.7 | 24. | III $7_1$ | 1.33 | 19.54 | 45. | I $11_1$ | 3.5 | 33.9 |
| 4. | P21 | 0.6 | 11.3 | 25. | III $8_1$ | 6.96 | 31.04 | 46. | I $14_1$ | 7.3 | 15.4 |
| 5. | P29 | 3.9 | 28.9 | 26. | III $8_3$ | 1.70 | 26.72 | 47. | 1 $1_1$ | 1.7 | 6.8 |
| 6. | M12 | 0 | 16.4 | 27. | I $14_1$ | 3.76 | 4.89 | 48. | I BH | 1.6 | 15.0 |
| 7. | P44 | 1.5 | 9.4 | 28. | III $2_2$ | 0 | 6.32 | 49. | I $1_5$ | 2.2 | 31.1 |
| 8. | P31 | 0 | 35.7 | 29. | III $6_3$ | 10.27 | 19.97 | 50. | I $8_2$ | 2.7 | 33.2 |
| 9. | P33 | trace | 5.9 | 30. | III $2_{21}$ | 0 | trace | 51. | I 9 | 3.9 | 39.2 |
| 10. | M5 | 0.9 | 10.4 | 31. | III $6_7$ | 2.04 | 17.55 | 52. | I 8 | 0.8 | 18.9 |
| 11. | II $6_2$ | 2.3 | 10.0 | 32. | II $5_1$ | 1.42 | 12.39 | 53. | II $4_3$ | 1.81 | 17.23 |
| 12. | II $6_5$ | 3.0 | 15.3 | 33. | III $11_2$ | 8.19 | 40.21 | 54. | I $7_3$ | 1.77 | 4.38 |
| 13. | II $6_6$ | 1.1 | 14.9 | 34. | K1 | 2.36 | 1.61 | 55. | I $1_6$ | 3.5 | 33.1 |
| 14. | II $5_1$ | 1.9 | 12.2 | 35. | K7 | 1.79 | trace | 56. | I 10 | 1.6 | 21.0 |
| 15. | T35 | 1.6 | 15.8 | 36. | I $14_1$ | 3 | 5.52 | 57. | I $1_9$ | 1.3 | 22.0 |
| 16. | 119 | 1.3 | 13.4 | 37. | 15 | 1.93 | 19.97 | 58. | I $7_2$ | 3.4 | 36.9 |
| 17. | I 13 | 3.74 | 6.8 | 38. | 16 | 0.86 | 23.94 | 59. | I $9_2$ | 2.5 | 6.02 |
| 18. | I $2_1$ | 0 | 9.5 | 39. | 19 | 1.34 | 9.75 | 60. | I $2_1$ | 4.8 | 7.1 |
| 19. | I $1_6$ | 3.3 | 5.7 | 40. | 22 | 0 | 10.18 | 61. | 1 $7_1$ | 2.7 | 27.4 |
| 20. | I $11_2$ | 3.12 | 3.8 | 41. | I S | 2.43 | 5.29 | 62. | I $11_2$ | 2.6 | 17.5 |
| 21. | I $10_1$ | 0.89 | 11.9 | 42. | Б$3_2$ | 1.64 | 12.09 | | | | |

As a result of this study, strains that are known to be good producers of intracellular and extracellular catalase were selected. The intracellular CAT activity of the strains studied ranged widely between 1.61 and 40.2 U/mg protein, 15 of which appeared to be good producers with activity above 20 U/mg protein (Figure 1).

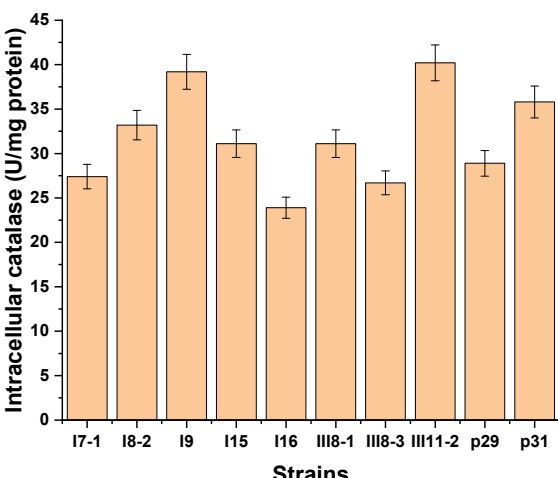

**Figure 1.** Strains good producers of intracellular catalase at 25 °C for 72 h.

The best production of intracellular catalase was shown by strains p29 (28.9 U/mg), III11-2 (40.2 U/mg), p31 (35.8 U/mg), I9 (39.2 U/mg), III8-1 (31.0 U/mg), I8-2 (33.2 U/mg), I7-1 (27.4 U/mg), III8-3 (26.7 U/mg), I15 (31.1 U/mg), and I16 (23.9 U/mg).

The investigated filamentous fungi belonged to different thermal classes: psychrophiles, psychrotolerants, and mesophiles. Analysis of the results did not reveal a clear relationship between catalase biosynthesis and the thermal preferences of the strains.

Extracellular enzymes are of great interest in terms of their application [45]. They have several advantages over intracellular ones: they are more economically viable and easier to

obtain in purified form. In the published scientific literature, reports of extracellular CAT production by microorganisms are very rare.

Our study revealed that almost all of the Antarctic strains investigated showed the ability to synthesize extracellular CAT, but its activity was lower compared to that of intracellular CAT and was in the range of 0.86–10.27 U/mg protein (Table 1). Strains showing a relatively good production ability are emerging (Figure 2). The selected strains showing higher extracellular catalase activity (above 3 U/mg protein) were p22 (3.6 U/mg), p29 (3.9 U/mg), I16 (3.3 U/mg), I9 (3.9 U/mg), I11-2 (2.6 U/mg), I13 (4.25 U/mg), I14-1 (7.3 U/mg), III2-1 (3.20 U/mg), III6-3 (10.27 U/mg), III8-1 (6.96 U/mg), and III11-2 (8.19 U/mg). Particularly good producers of extracellular CAT were strains III8-1, III11-2, and III6-3 with activities of 7, 8.2, and 10.3 U/mg proteins, respectively (Figure 2).

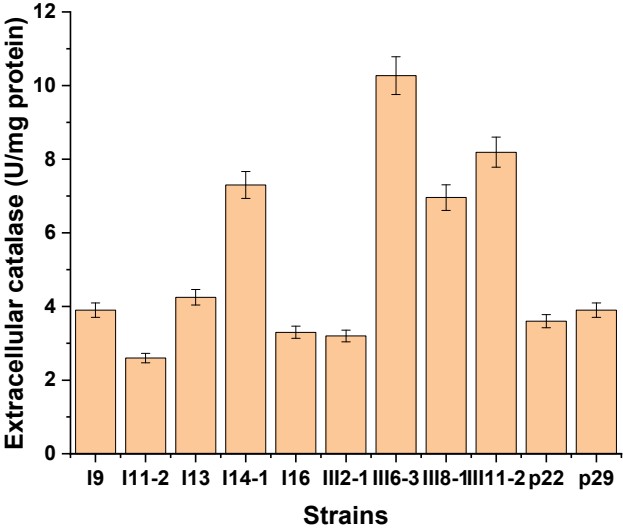

**Figure 2.** Strains showing good extracellular catalase activity at 25 °C for 72 h.

### 3.2. Identification of Extracellular Catalase-Producing Strains

The selected extracellular catalase-producing strains were identified to the genus level using the classical taxonomic approach. They all belonged to Division Ascomycota.

The identification to species level was performed using molecular methods based on the conservativeness of the small RNA subunit as a taxonomic characteristic.

The percentage similarity found with the species reported in NCBI is presented in Table 2.

**Table 2.** Molecular genetic identification of the strains studied.

| № | Strain Abbreviation | Strain Name | Macroscopic Picture | Taxonomy | NCBI GenBank Database Accession Numbers |
|---|---|---|---|---|---|
| 1. | P29 | *Penicillium griseofulvum* | | Class: Eurotiomycetes Order: Eurotiales Family: *Trichocomaceae* Genus: *Penicillium* | MT722118 |
| 2. | P22 | *Penicillium* sp. | | Class: Eurotiomycetes Order: Eurotiales Family: *Trichocomaceae* Genus: *Penicillium* | MT730073 |

**Table 2.** *Cont.*

| № | Strain Abbreviation | Strain Name | Macroscopic Picture | Taxonomy | NCBI GenBank Database Accession Numbers |
|---|---|---|---|---|---|
| 3. | I1-6 | *Penicillium rubens* |  | Class: Eurotiomycetes<br>Order: Eurotiales<br>Family: *Trichocomaceae*<br>Genus: *Penicillium* | MT758185 |
| 4. | I9 | *Aspergillus fumigatus* |  | Class: Eurotiomycetes<br>Order: Eurotiales<br>Family: *Aspergillaceae*<br>Genus: *Aspergillus* | MT758189 |
| 5. | I11-2 | *Penicillium rubens* |  | Class: Eurotiomycetes<br>Order: Eurotiales<br>Family: *Trichocomaceae*<br>Genus: *Penicillium* | MT722143 |
| 6. | I13 | *Penicillium commune* |  | Class: Eurotiomycetes<br>Order: Eurotiales<br>Family: *Trichocomaceae*<br>Genus: *Penicillium* | MT730061 |
| 7. | I14-1 | *Pseudogymnoascus pannorum* |  | Class: Leotiomycetes<br>Order: Helotiales<br>Family: *Myxotrichaceae*<br>Genus: *Geomyces* | MT722119 |
| 8. | III2-1 | *Epicoccum nigrum* |  | Class: Dothideomycetes<br>Order: Pleosporales<br>Family: *Didymellaceae*<br>Genus: *Epicoccum* | OR844311 |
| 9. | III6-3 | *Aspergillus glaucus* |  | Class: Eurotiomycetes<br>Order: Eurotiales<br>Family: *Aspergillaceae*<br>Genus: *Aspergillus* | JN206683.1 |
| 10. | III8-1 | *Penicillium commune* |  | Class: Eurotiomycetes<br>Order: Eurotiales<br>Family: *Trichocomaceae*<br>Genus: *Penicillium* | MT722126 |
| 11. | III11-2 | *Penicillium rubens* |  | Class: Eurotiomycetes<br>Order: Eurotiales<br>Family: *Trichocomaceae*<br>Genus: *Penicillium* | OR844310 |

### 3.3. Selection of an Extracellular Catalase-Producing Strain

The time course of the extracellular catalase synthesis in the selected strains was studied (Figure 3). Figure 3 shows that there is no clear tendency of the dependence of maximum enzyme activity on the cultivation time. In five of the strains, the enzyme maximum was reached at 120 h from the beginning of cultivation, in four at 96 h, and in two at 72 h. Of all selected strains, strain III11-2- *Penicillium rubens* appeared to have the best extracellular enzyme production ability and was therefore selected for further experiments.

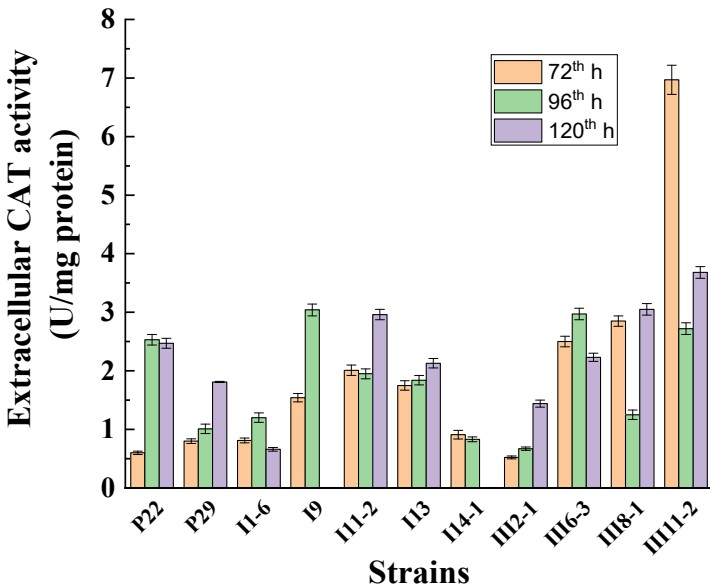

**Figure 3.** Time course of extracellular catalase production by selected Antarctic strains cultivated at 25 °C.

The strain grew well on three agar media (PDA, SA, and CzDA) in a temperature range characteristic of mesophilic strains (between 20 and 40 °C) (Figure 4). Macroscopic observations showed that the single-spore colony of the producer had abundant and velvety aerial mycelia with a green color, yellow exudates, a white margin without fruiting bodies and spores (Figure 4).

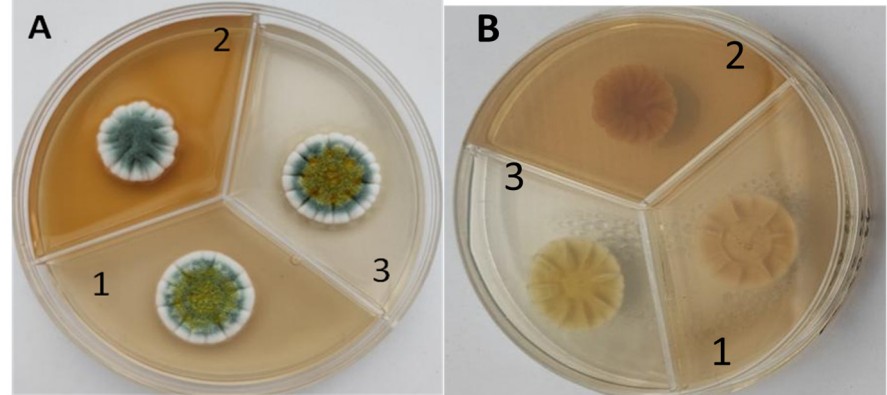

**Figure 4.** Macroscopic photos of the model strain cultured on three agar media (1—PDA, 2—SA, and 3—CzDA); (**A**) front side, (**B**) reverse side.

### 3.4. Investigation of the Role of Extracellular Catalase under Normoxic Conditions and under Low-Temperature Stress

One of the aims of our study was to investigate the role of extracellular catalase in oxidative stress conditions caused by low temperatures. The development of *Penicillium*

*rubens* III11-2 was studied under submerged cultivation in a liquid culture medium at the optimum growth temperature (25 °C) and under low-temperature stress conditions (15 °C).

As illustrated in Figure 5A, the highest amount of biomass was formed at a growth temperature of 25 °C, which confirmed the affiliation of the strain to mesophilic fungi. At the lower temperature used, a slower development of the model strain was observed. The carbon source uptake results were consistent with the development of the strain at the two temperatures used in the study. Cultivation of *Penicillium rubens* III11-2 at 25 °C resulted in rapid consumption of the carbon source while a decrease in the cultivation temperature slowed glucose consumption (Figure 5B).

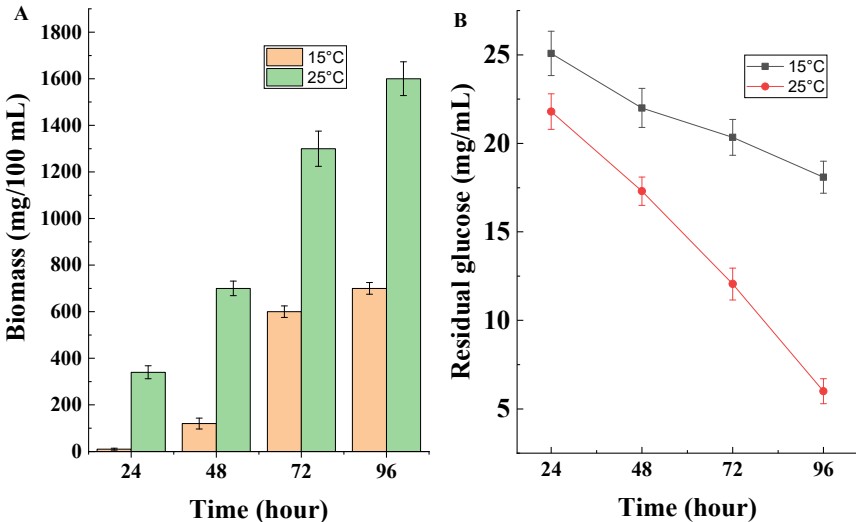

**Figure 5.** (**A**) Biomass formation and (**B**) glucose consumption of the model strain cultured at optimum temperature and under conditions of low-temperature stress.

Fungi, like all eukaryotic organisms, possess a survival strategy in the face of stress caused by various stimuli. The first line of antioxidant defense against the damaging action of free oxyradicals includes the enzymes superoxide dismutase and catalase. Our results showed an increase in total enzyme activity as a result of low-temperature exposure (Figure 6A,B). There was a clear trend of increased extracellular SOD activity at all time intervals studied, with a 1.64-fold increase at the 96th hour of cultivation compared to the enzyme activity when the strain was cultured at 25 °C (Figure 6A). A similar trend was observed for the activity of the second antioxidant enzyme, CAT (Figure 6B).

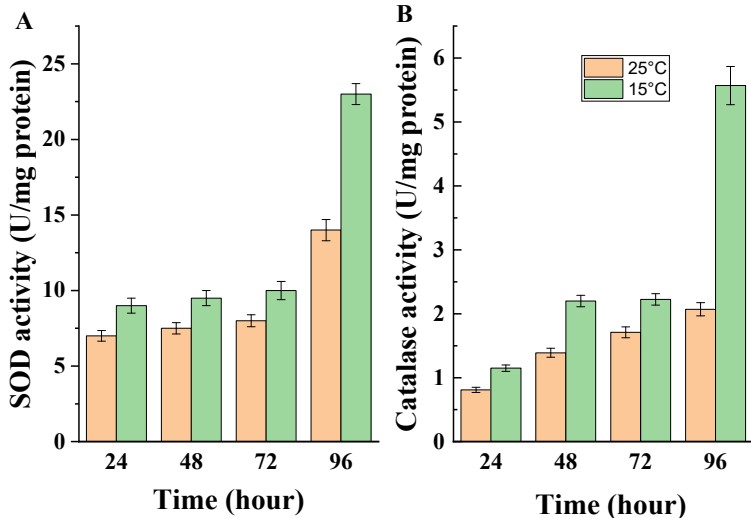

**Figure 6.** (**A**) Extracellular SOD and (**B**) extracellular CAT activity of the model strain cultured at different temperatures.

The cells of the model strain demonstrated increased extracellular CAT synthesis under conditions of low-temperature stress at all periods examined. At the 96th hour after the start of cultivation, this activity exceeded, 2.69-fold, the activity of the enzyme synthesized at the optimal temperature for the strain (Figure 6B).

*3.5. Purification and Characterization of CA CAT*

3.5.1. Preparation of Purified Enzyme

In the present study, the culture filtrate (315 mL) obtained from submerged fungal culture was used in the experiments. The purification protocol included concentration by ultrafiltration, ammonium sulphate precipitation, and ion-exchange chromatography on Q-Sepharose to produce a homogenous enzyme. Purification procedures are summarized in Table 3 and Figure 7. The ultrafiltration and ammonium sulphate precipitation at 40% saturation yielded 23% activity, 18.31 U/mg protein specific activity, and 3.5-fold purification. Further purification was achieved using ion-exchange chromatography on Q-Sepharose resulting in a single active peak (Figure 7) with specific activity of 513.72 U/mg protein, which was 98.4-fold higher than that of the crude enzyme with a 19.97% yield.

**Table 3.** Purification scheme of extracellular catalase by *P. rubens* III 11-2.

| Sample | Volume (mL) | Protein (mg) | Specific CAT Activity (U/mg) | Total CAT Activity (U) | Yield (%) | Degree of Purification (Fold) |
|---|---|---|---|---|---|---|
| Output | 315 | 1.466 | 5.22 | 2410.50 | 100 | 1 |
| Ultrafiltration | 24 | 2.465 | 6.02 | 356.14 | 14.77 | 1.15 |
| $(NH_4)_2SO_4$ Saturation | 20 | 1.528 | 18.31 | 559.55 | 23.21 | 3.51 |
| Column Q-Sepharose | 7.5 | 0.125 | 513.72 | 481.60 | 19.97 | 98.4 |

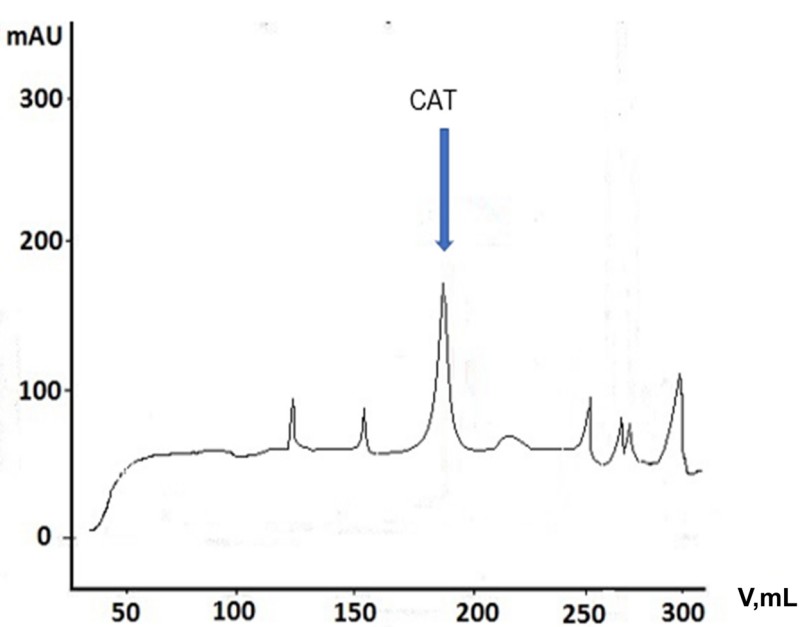

**Figure 7.** Resulting fractions after purification of CAT using FPLC system.

In this case, the impurities passed through the column and the enzyme bound to the hydrophobic medium in the column. The very distinct peak was evidence of the presence of the enzyme (Figure 7, Table 3). CAT was eluted with 0.1 M potassium phosphate buffer at pH 7.5. Next, the resulting enzyme was concentrated using centrifuge tubes with a filter permeability of 10 kDa.

This purification approach proved to be very successful since, under the same chromatographic conditions, a successful separation of the enzyme and the remaining components was achieved. The large volume of the applied fraction (10–12 mL) for purification

and the high elution rate (70 mL/h) from the columns were advantages of the process. The fractions corresponding to the peaks in the figure were assayed for enzyme activity. The highest peak in the figure corresponded to the catalase enzyme activity.

As seen in Table 3, the scheme developed in this study was extremely effective. After the Q Sepharose column, 98.4-fold purification of the enzyme was achieved with a 19.97% yield.

The purity of the resulting active fraction was checked using SDS-PAGE. In Figure 8A, a streak (marked in a circle) of the purified enzyme fraction is visualized. The presence of the enzyme was visualized using native electrophoresis (Figure 8B). In accordance with the molecular standard, we assumed 42 kDa for one subunit.

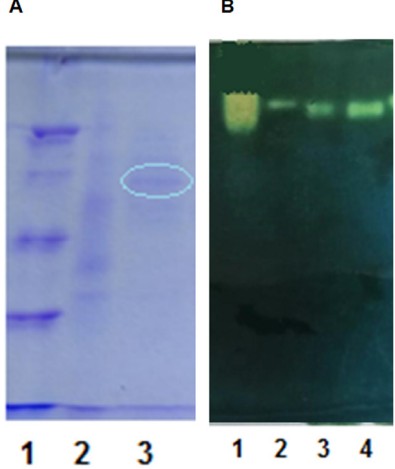

**Figure 8.** Electrophoretic visualization of the purified enzyme CAT. (**A**) SDS PAGE; (1) protein standard; (2) initial sample—culture fluid after mycelial filtration; (3) extracellular catalase—sample peak after partial purification (in circle). (**B**) Native PAGE standard catalase from *Aspergillus niger* (Sigma); (2) initial sample—culture fluid after mycelial filtration; (3) aggregate sample—after pooling fractions with molecular mass above 10 KDa; (4) peak sample after partial purification.

### 3.5.2. Properties of the Purified Enzyme Preparation

*Effect of temperature on catalase enzyme activity*

The effect of temperature on the activity and stability of purified CAT enzyme preparation from *Penicillium rubens* III11-2 was determined over a wide temperature range from 5 °C to 80 °C (Figure 9).

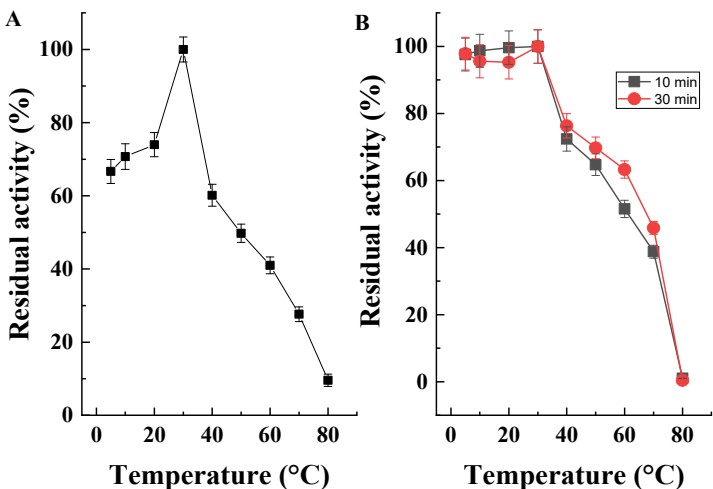

**Figure 9.** (**A**) Effect of temperature on the activity of purified CAT and (**B**) temperature stability of the purified enzyme when exposed to different temperatures for 10 min (■) and 30 min (●).

The optimum temperature for the reaction was 30 °C and a high level of activity (>70%) was maintained between 5 °C and 40 °C. At 50, 60, and 70 °C, the enzyme retained 50%, 41%, and 28% of its initial activity, respectively. The purified enzyme was stable at 30 °C for 10 and 30 min and maintained about 75%, 65%, 55%, and 40% activity at 40, 50, 60 and 70 °C. At 80 °C, the residual activity was about 1% (Figure 9B).

## 4. Discussion

Our results indicate that Antarctic filamentous fungi belonging to different thermal classes demonstrated significant biotechnological potential for catalase synthesis. Some of them possess the ability to produce a rare enzyme, extracellular CAT. As a result of screening for catalase producers, we selected the strain *Penicillium rubens* III11-2 as an efficient extracellular catalase producer, belonging to mesophilic fungi.

Data on extracellular catalase production have been published for *Claviceps purpurea* [46] and *Aspergillus foetidus* [15,47]. The authors suggested that the production of extracellular catalase was induced when rice plants were infected with the fungus, and it played a role in the pathogenesis of ergot diseases. In a study by Russian scientists, synthesis of extracellular catalase was also demonstrated by filamentous fungi of the genera *Penicillium*, *Talaromyces*, and *Aspergillus* [48]. The Kacem-Chaouche group investigated the synthesis and secretion of extracellular CAT in fungi [15]. They suggest that there is a correlation between the synthesis, secretion and morphology of hyphae in the outermost layer of mycelial pellets, and it is related to the apical and subapical region, the so-called active region in deep cultivation [49]. Fenice et al., 1997, and Laura Zucconi et al., 2020, published information on extracellular enzymes produced by Antarctic filamentous fungi [45,50].

Our previous studies have provided information on the importance of catalase for the survival of Antarctic filamentous fungi under low-temperature stress conditions, which, however, concern the intracellular enzyme [51–53]. Our present study is one of the first to investigate extracellular catalase and its role under cold stress conditions in Antarctic filamentous fungi.

The antioxidant enzyme defense system is a very essential mechanism for the survival of filamentous fungi under abiotic stress. Cellular antioxidant enzyme defenses, SOD and CAT, have been associated with micromycetes resistance to various stress factors [12,54]. Both antioxidant enzymes degrade $^\bullet O_2^-$ and $H_2O_2$, respectively, the generation of which is accelerated under stress conditions. Increased activity of antioxidant enzymatic defense is one of the main characteristics of stress caused by low temperatures [33,55]. ROS plays a role in cell signaling. They can serve as both intra- and intercellular messengers in apoptosis, gene expression, and activation of intracellular signaling cascades. To eliminate the harmful effects of ROS, cells have developed a range of defense mechanisms, such as the activation of several enzymes. Superoxide dismutase, peroxidase, and catalase are the most prevalent and vital enzymes found in nearly all living organisms. Our results confirm existing data in the literature and extend them to filamentous fungi. This is also new information on Antarctic strains. SOD activity showed an increase when the strain was cultured at 15 °C for all periods monitored (Figure 6A).

As can be seen in Figure 6B, the trend of catalase level change is identical to that of SOD. The dismutation of superoxide radicals resulted in the formation of $H_2O_2$, which must be neutralized by catalase or peroxidase [56]. This, in turn, requires higher activity of the second enzyme of the antioxidant defense.

The results of our study confirm the essential role of extracellular antioxidant enzymes under conditions of low-temperature stress. The organism responds with increased antioxidant protection to immobilize the large amount of free oxyradicals [57].

The extracellular antioxidant enzymes SOD and CAT are involved in the response of filamentous fungi against low-temperature stress. Extracellular catalase plays a major role in their survival under such conditions.

In the course of our study, an efficient purification scheme for extracellular catalase produced by the Antarctic strain *Penicillium rubens* III11-2 was developed. The purification

protocol is simple and enzyme purification is achieved in a single chromatographic step. With the application of ion exchange chromatography on Q-Sepharose, CAT with high specific activity (513 U/mg protein) was obtained with a yield of 19.97% and a purification rate of 98.4-fold. It should be noted that information on the purification of extracellular catalase is rare. Similar results were reported by Kacem-Chaouche et al. (2013) for *A. foetidus* using two-step chromatography [47]. Rogalski et al. (1998) proposed a purification procedure of extracellular CAT from *Aspergillus niger* in three chromatographic steps (DEAE Sepharose, Phenyl-Sepharose and chromatofocusing on PBE 94 column) [58].

Since data on the purification of extracellular catalase are very scarce, we can compare the results obtained in this work with other extracellular enzymes from filamentous fungi. Adeoyo et al. (2018) achieved a 27% yield and 26-fold purification of amyloglucosidase from the mycorrhizal fungus *Leohumicola incrustata* [59]. The *Mycena purpureofusca* laccase purification scheme allows a 9% yield with 26-fold purification [60]. It should be noted that extracellular catalase from *P. rubens* strain III11-2 is the first purified enzyme of Antarctic origin. Fiedurek et al. (2003) reported extracellular catalase from strains isolated from the Arctic, but the authors did not provide information on the purification of the enzyme [61]. Some specific characteristics of the resulting enzyme, such as temperature optimum and temperature stability, were investigated. The results showed that incubation of the enzyme at different temperatures preserved its activity in the optimum temperature range for 30 min and showed stability in the low-temperature range. As the temperature increased, significant instability of the specific activity was observed. The resulting purified enzyme preparation exhibited maximum enzymatic activity in the temperature range of 5–40 °C and temperature stability between 0 and 30 °C. CA enzymes exhibit maximum specific activity at temperatures below 40 °C, in contrast to mesophilic (50–60 °C) or thermophilic (80° and higher) [27]. Sensitivity at temperatures above 40 °C is the main characteristic of CA enzymes. These results characterized the enzyme from *P. rubens* III11-2 as a CA enzyme. The enzymatic activity of CA extracellular *P. rubens* CAT characterizes it as an enzyme with great potential for use in processes needing low temperatures for application. At the same time, most commercial catalases have optimal activity at between 20 °C and 50 °C, which makes them unable to withstand such adverse conditions [61].

According to Margesin and Schinner (1999), CA enzymes have optimum temperatures around 30 °C, and they retain their high activity even at 0 °C [62]. The majority of microorganisms that are the subject of scientific reports on CA enzymes are bacteria. The first characterized CA catalase was produced by the facultative psychrophilic bacterium *Vibrio rumoiensis* S-1T. Compared to the mesophilic catalase from *Micrococcus luteus* and from bovine liver, this enzyme is noticeably less thermostable [28]. Wang et al. (2008) reported a cold-adapted catalase from *Bacillus* sp. N2a with a low optimal temperature of 25 °C [31]. Cat-2 from the mesophilic *Serratia marcescens* SYBC-01 isolated from Wuxi City, China, exhibits significant relative activity (73.8%) at 0 °C, and its ideal temperature is roughly 20 °C. The authors characterized this enzyme as a type of cold-adaptive enzyme [30]. Krumova et al. (2021) reported CA CAT produced by a psychrotolerant strain of *P. griseofulvum* [33]. The catalase studied was an intracellular enzyme. There are few reports on similar extracellular enzymes.

Based on the published studies, it is known that CA catalases generally show higher instability at moderately high temperatures compared to their mesophilic counterparts. However, studies on the temperature stability of CA CAT produced by *Serratia marcescens* SYBC-01 showed that 75% of its initial activity was retained after incubation at 60 °C for 60 min. All of the above indicate higher thermal stability than three other cold-adapted catalases and some mesophilic catalases [30]. Our results also showed the higher stability of the extracellular catalase studied.

## 5. Conclusions

In conclusion, the Antarctic isolates of filamentous fungi are good producers of the rare enzyme extracellular catalase. This enzyme is involved in the cellular response against

low-temperature stress, which supports the survival of those fungi in extremely cold habitats. The strain *Penicillium rubens* III11-2, belonging to the mesophilic fungi, proved to be an efficient producer of extracellular catalase. The purified enzyme preparation exhibited maximum activity at low temperatures (5–30 °C) and low thermostability, which characterizes it as a CA enzyme. To our knowledge, this is the first purified extracellular CA catalase preparation from Antarctic filamentous fungi.

**Author Contributions:** Conceptualization, E.K.; methodology, R.A., G.S. and B.S.; software, M.A.; validation, E.K. and G.S.; formal analysis, J.M.-S.; investigation, G.S., V.D., L.Y. and Z.K.; data curation, Z.K. and G.S.; writing—original draft preparation, E.K.; writing—review and editing, E.K. and M.A.; visualization, J.M.-S. and Z.K. All authors have read and agreed to the published version of the manuscript.

**Funding:** This research received no external funding.

**Data Availability Statement:** The raw data supporting the conclusions of this article will be made available by the authors on request.

**Conflicts of Interest:** The authors declare no conflicts of interest.

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
