# Peer review of "A Novel Extracellular Catalase Produced by the Antarctic Filamentous Fungus Penicillium Rubens III11-2"

_fermentation, doi:10.3390/fermentation10010058_

Round 1
Reviewer 1 Report
Comments and Suggestions for Authors
The work entitled: “Novel Extracellular Catalase by Antarctic Filamentous Fungus Penicillium Rubens III11-2”, is interesting, however it requires several changes and improving the description of the results, as well as other sections. Furthermore, the authors do not indicate the number of replicates, nor do they perform statistical analysis.
In the abstract, the authors should address more about the methodological strategy of enzyme purification.
2. Materials and Methods
- Improve the wording of the following sentence: A 200 mg 48h mycelium was used…
- Visualization of isolated DNA by 1% agarose gel. DNA visualization is carried out with ethidium bromide or another type of intercalant. Improve writing.
Visualization of PCR products (about 0.7 kb per ITS) through 1% agarose gel. This phrase can be combined with the previous one, it talks about the same thing.
- PCR reaction. Universal primers for the ITS region (ITS1: 5'-TCCGTAGGTGAACCTGCG-3' and ITS4: 5'-TCCTCCGCTTATTGATATATGC-3) were used. The primers were designed by the authors or taken from some reference, if so, add reference.
- Sequencing of ITS regions at Macrogen Inc. (Amsterdam, The Netherlands). Improve writing.
-The authors do not describe how they obtained the intracellular extracts
Results
- Furthermore, it does not make sense to place the same values in tables and then in graphs, only that the graphs indicate the error bars, but they do not mention the experimental replicates either.
- In the results section (Fig. 2) it is very clear that strain III6-3 produces more extracellular CAT, however in the abstract the authors indicate that strain III11-2 is the best. Clarify this point.
- At what time were the extracellular activities reported in Table 1 (or Fig. 2) quantified. This is due to the fact that strain III11-2 has lower activity, even with what was reported in strain III6-3. Please explain.
- Fig. 3. How many replicates did the authors perform? The error bars correspond to the standard deviation
- Figures 5 and 6 can be unified and placed as A and B, respectively.
- Same comment as the previous one for figures 7 and 8.
- The extracellular CAT activity in figure 3 is approximately 7 at 72 h, and now in figure 8 it is approximately 2 at that time, and greater activity is presented at 96 h. Why does this happen?
- Figure 9 does not represent anything different from what is described in the text. It can be removed.
- The black line in Figure 10 indicates the fractions or the activity, the graph should represent both
- Change SDS-RAGE to SDS-PAGE.
- Figure 11A, lane 1, place the corresponding sizes in kDa on one side of the gel.
- The authors do not mention the approximate molecular mass of the purified catalase, or why did they not determine it?
- Figure 11 is very dense, the text can be placed in the text of figure 11. By the way, figure 11B does not describe the conditions to visualize catalase, it is important to describe the methodology.
- Figures 12 and 13 can be unified and placed as A and B, respectively.
- It would be important for the authors to sequence the purified protein to verify that it is a catalase.
4. Discussion
Change E/mg protein to U/mg protein
The authors do not talk about the molecular masses reported in other organisms, to take as a reference with that obtained in this work.
Comments on the Quality of English LanguageModerate editing of English language required.
In the comments to the authors, spelling and writing errors were indicated.
Author Response
Please, see the attachment!

Reviewer 2 Report
Comments and Suggestions for Authors
The manuscript entitle “ INVESTIGATION OF NOVEL EXTRACELLULAR CATALASE BY ANTARCTIC FILAMENTOUS FUNGUS PENICILLIUM RUBENS III11-2” by Koleva at al deals with a very interesting fact of survival of Antarctic filamentous fungi Penicillium by catalase enzyme production. The methods are sound, results are nice; however the overall representation of the manuscript is very poor which must be corrected. For example:
1. All the column graphs in this paper are very poor representations. The authors are suggested to revise it.
2. The Figure 9 of enzyme production protocol is more fit in methodology which also need to be better represented and not as it is now.
3. Figure 11 is also another example of poor image formation.
4. The conclusion are better presented more concise and one paragraph.
5. The FPLC detailed methodology is missing in the manuscript.
Comments on the Quality of English Language
Moderate language revision throughout the manuscript is needed. The manuscript cannot to be acccepted in its present format and hence needs careful revision.
Author Response
- The manuscript entitle “ INVESTIGATION OF NOVEL EXTRACELLULAR CATALASE BY ANTARCTIC FILAMENTOUS FUNGUS PENICILLIUM RUBENS III11-2” by Koleva at al deals with a very interesting fact of survival of Antarctic filamentous fungi Penicillium by catalase enzyme production. The methods are sound, results are nice; however the overall representation of the manuscript is very poor which must be corrected. For example:
- All the column graphs in this paper are very poor representations. The authors are suggested to revise it.
Reply: We improved the graphs.
- The Figure 9 of enzyme production protocol is more fit in methodology which also need to be better represented and not as it is now.
Reply: We agree with the reviewer's suggestion and we removed figure 9.
- Figure 11 is also another example of poor image formation.
Reply: Thank you for the suggestion. We improved the figure.
- The conclusion are better presented more concise and one paragraph.
Reply: We presented the conclusion in one paragraph.
- The FPLC detailed methodology is missing in the manuscript.
Reply: We added the FPLC detailed methodology
- Moderate language revision throughout the manuscript is needed. The manuscript cannot to be acccepted in its present format and hence needs careful revision.
Reply: The English text of the corrected version has been revised by Mrs A. Nikolaeva, MA in English language and literature
Reviewer 3 Report
Comments and Suggestions for Authors
This paper described an extracellular catalase from filamentous fungus Penicillium rubens isolated from Antarctic region.
Major:
1. Although most of the presented data seemed to be preliminary, the catalase that exhibited distinct temperature dependency is quite interesting. Because the reaction is very fast in spectrophotometric measurement and the enzymatic reaction itself is a redox reaction in which hydrogen peroxide passes through the hydrophobic channels of the enzyme, it is thought that it is less susceptible to the effects of molecular movement depend on the temperature. If possible, in order to show the validity of the experimental method, please present the temperature dependence of commercially available catalase(s) at the same time.
2. Please discuss deeply about relationships among temperature, incubation period, cold stress, and induction of extracellular enzyme.
3. Please discuss most appearing point at first paragraph in Discussion.
4. Please describe methods for statistical processing of data in “Materials and Methods”.
Minor:
1) Figure 1 & 2: Please describe temperature and period for the cultivation in the legend.
2) Figure 3: Please describe incubation temperature for the cultivation.
3) Figure 4: It is difficult to understand which medium each is. Please describe temperature and period for the cultivation in the legend.
4) Figure 10: It is difficult to understand how to monitor CAT. Please explain in the legend.
5) Table 2, No.2: “sp.” in “Penicillium sp.” need not to be italics.
6) Table 3: Saturation → (NHâ‚„)â‚‚SOâ‚„ saturation; Sepharose → Q-Sepharose
7) Figure: Please illustrate the location of the anode and cathode and the direction of electrophoresis.
8) P14 L7: 513 E → 513 U
9) P14 L21 from the bottom: a cold-active (CA) enzyme → a CA enzyme
10) “sp.” in “Bacillus sp.” need not to be italics.
Comments on the Quality of English LanguageNo comment.
Author Response
This paper described an extracellular catalase from the filamentous fungus Penicillium rubens isolated from the Antarctic region.
Major:
- Although most of the presented data seemed to be preliminary, the catalase that exhibited distinct temperature dependency is quite interesting. Because the reaction is very fast in spectrophotometric measurement and the enzymatic reaction itself is a redox reaction in which hydrogen peroxide passes through the hydrophobic channels of the enzyme, it is thought that it is less susceptible to the effects of molecular movement depend on the temperature. If possible, in order to show the validity of the experimental method, please present the temperature dependence of commercially available catalase(s) at the same time.
Reply: We did not find commercially available cold-active catalase to make a comparison with ours.
- Please discuss deeply about relationships among temperature, incubation period, cold stress, and induction of extracellular enzyme.
Reply: These relationships are discussed in our previous paper we refereed.
- Please discuss most appearing point at first paragraph in Discussion.
Reply: The first paragraph in the Discussion is based on our experiments concerning the present manuscript.
- Please describe methods for statistical processing of data in “Materials and Methods”.
Reply: Information about the statistical evaluation of the results was added to the section Material&Methods.
Minor:
- Figure 1 & 2: Please describe temperature and period for the cultivation in the legend.
Reply: We improved the legend of the figures 1 and 2.
- Figure 3: Please describe incubation temperature for the cultivation.
Reply: We added incubation temperature for the cultivation for Fig. 3.
- Figure 4: It is difficult to understand which medium each is. Please describe temperature and period for the cultivation in the legend.
Reply: We improved the figure in accordance with to reviewer's suggestions.
- Figure 10: It is difficult to understand how to monitor CAT. Please explain in the legend.
Reply: The graph in the figure indicates the fractions. Each fraction was analyzed for enzyme activity.
- Table 2, No.2: “sp.” in “Penicillium sp.” need not to be italics.
Reply: We improved it.
6) Table 3: Saturation → (NHâ‚„)â‚‚SOâ‚„ saturation; Sepharose → Q-Sepharose
Reply: We improved the abbreviations.
- Figure: Please illustrate the location of the anode and cathode and the direction of electrophoresis.
Reply: We improved the figure.
- P14 L7: 513 E → 513 U
Reply: We improved it.
- P14 L21 from the bottom: a cold-active (CA) enzyme → a CA enzyme
Reply: We improved it.
- “sp.” in “Bacillus sp.” need not to be italics.
Reply: We improved it.
Round 2
Reviewer 1 Report
Comments and Suggestions for Authors
The work entitled: “Novel Extracellular Catalase by Antarctic Filamentous Fungus Penicillium Rubens III11-2”.
The revision was complicated because the authors did not eliminate text and figure changes in the new version of the manuscript.
The authors answered the requested questions and appropriately modified the manuscript. After making minor changes, this new version can be considered for publication in this journal.
Minor comments
Penicillium Rubens can be considered as a keyword
All scientific names of species such as Vibrio rumoiensis, V. salmonicida, Serratia marcescens, Bacillus subtilis, etc., which appear in the introduction as well as in all sections of the manuscript, should be placed in italics.
Section 2.3: Define NBT
It is suggested to change Molecular genetic identification to Molecular identification.
Section 3.2: “The conformity of the sequences we obtained with the sequences of collection strains from different species in NCBI is presented in Table 2”. This paragraph is not clear. The authors must indicate in the text (or table 2) the % similarity found with the species reported in NCBI.
Figure 4 is not clear in the new version. The authors show the same Petri dish on both sides. If so, describe it in the text in figure 4.
Comments on the Quality of English LanguageMinor editing of English language required
Author Response
Dear Reviewer, thank you for your thorough evaluations and insightful recommendations. We believe that the adjustments made after your advices have significantly improved the manuscript.
Below we provide the point-by-point responses.
- The revision was complicated because the authors did not eliminate text and figure changes in the new version of the manuscript.
Reply: Sorry about this. We prepared the new version of the manuscript.
The authors answered the requested questions and appropriately modified the manuscript. After making minor changes, this new version can be considered for publication in this journal.
- Penicillium rubens can be considered as a keyword
Reply: We added Penicillium rubens to the keywords
- All scientific names of species such as Vibrio rumoiensis, V. salmonicida, Serratia marcescens, Bacillus subtilis, etc., which appear in the introduction as well as in all sections of the manuscript, should be placed in italics.
Reply: Sorry for miss! We placed all scientific names of species in all sections of the manuscript in italic.
- Section 2.3: Define NBT
Reply: Sorry for miss! Nitro blue tetrazolium chloride (NBT). We added it in Material and Methods section.
- It is suggested to change Molecular genetic identification to Molecular identification
Reply: We changed it in accordance with your recommendation.
- Section 3.2: “The conformity of the sequences we obtained with the sequences of collection strains from different species in NCBI is presented in Table 2”. This paragraph is not clear. The authors must indicate in the text (or table 2) the % similarity found with the species reported in NCBI.
Reply: We precisized the sentence in accordance with your advice.
- Figure 4 is not clear in the new version. The authors show the same Petri dish on both sides. If so, describe it in the text in figure 4.
Reply: We improved the figure 4 and its legend.
Reviewer 2 Report
Comments and Suggestions for Authors
The revision made by the authors with the reviewer’s comments looks satisfactory.
Comments on the Quality of English Language
Only minor changes and thorough language revision is needed now.
Author Response
Dear Reviewer, thank you for your thorough evaluations and insightful recommendations. We believe that the adjustments made after your advices have significantly improved the manuscript.
- The revision made by the authors with the reviewer’s comments looks satisfactory. Only minor changes and thorough language revision is needed now.
Reply: Thank you! We made some corrections in the manuscript to improve its quality.
Reviewer 3 Report
Comments and Suggestions for Authors
The manuscript described on extracellular cold-adapted catalase produced from Antarctic filamentous fungus Penicillium rubens III11-2.
Because the reaction is very fast in spectrophotometric measurement and the enzymatic reaction itself is a redox reaction in which hydrogen peroxide passes through the hydrophobic channels of the enzyme to react heme in the molecule. Therefore, it is thought that it is less susceptible to the effects of molecular movement depend on the temperature. If possible, in order to show the validity of the experimental method, please present the temperature dependence of commercially available mesophilic catalase(s) such as bovine catalase at the same time. Although the true intention of this review was not properly understood by the authors, the temperature stability showed that approximately 30% of the activity was lost in 10 minutes at 40°C, suggesting that the temperature dependence of activity was measured correctly.
Author Response
Dear Reviewer, thank you for your thorough evaluations and insightful recommendations. We believe that the adjustments made after your advices have significantly improved our manuscript.
The manuscript described on extracellular cold-adapted catalase produced from Antarctic filamentous fungus Penicillium rubens III11-2.
Because the reaction is very fast in spectrophotometric measurement and the enzymatic reaction itself is a redox reaction in which hydrogen peroxide passes through the hydrophobic channels of the enzyme to react heme in the molecule. Therefore, it is thought that it is less susceptible to the effects of molecular movement depend on the temperature. If possible, in order to show the validity of the experimental method, please present the temperature dependence of commercially available mesophilic catalase(s) such as bovine catalase at the same time. Although the true intention of this review was not properly understood by the authors, the temperature stability showed that approximately 30% of the activity was lost in 10 minutes at 40°C, suggesting that the temperature dependence of activity was measured correctly.
Reply: In accordance with the your remark, we performed an extensive review of the scientific and patent literature on the temperature dependence of commercial catalase. Data have been published on the thermostability of the following catalases:
- Yumoto et al. (2000) compared the effect of temperature on the activity of bovine liver catalase and that from Vibrio rumoiensis. The authors noted that the temperature optimum of bovine liver catalase was 40-60°C. [Isao Yumoto, Daisen Ichihashi, Hideaki Iwata, Anita Istokovics, Nobutoshi Ichise, Matsuyama, Hidetoshi Okuyama, Kosei Kawasaki Purification and Characterization of a Catalase from the Facultatively Psychrophilic Bacterium Vibrio rumoiensisS-1T Exhibiting High Catalase ActivityJ Bacteriol. 2000 Apr; 182(7): 1903–1909. doi: 1128/jb.182.7.1903-1909.2000]
- MiÅ‚ek et al. (2014) studied temperature dependence of the commercial catalase Terminox Ultra. This catalase kept its activity at temperature 35°C for about 30 hours; at 40 and 45°C for about 5 hours. [J. MiÅ‚ek , M. Wójcik , W. Verschelde Thermal stability for the effective use of commercial catalase Polish Journal of Chemical Technology, 16, 4, 75 — 79, 10.2478/pjct-2014-0073]
- MiÅ‚ek and WóJcik (2011) showed that the commercial catalase Terminox Ultra could be used for H2O2 decomposition at temperature range between 35-50°C. [Justyna MiÅ‚ek, Marek WóJcik Effect of temperature on the decomposition of hydrogenperoxide by catalase Terminox Ultra Przemyst Chemiczny, 2011, 90/6, 1260-1263]
- Christensen et al. (1992) get a patent for catalase production from strains of Scytalidium and Humicola, that kept 75% residual activity after 20 minutes at 70°C. [Bjørn Eggert Christensen, Niels Krebs Lange, Kousaku Daimon (1992) Catalase, its production and use. PATENT WO1992017571A1 Application PCT/DK1992/000098 events1992-03-27]
Taken together, according to Spiro and Griffith, 1997, the most commercial catalases have optimum activity at 20°C to 50°C and at neutral pH, making them unable to withstand the adverse conditions in some industrial processes. [Spiro, M.C. and Griffith, W.P. 1997. The mechanism of hydrogen peroxide bleaching. Textile Chemistand Colorist, 29: 12–13]
We added in the Discussion section the text follows:
The enzymatic activity of CA extracellular P. rubens CAT characterizes it as enzyme with great potential for use in processes need low temperatures for application. At the same time most commercial catalases have optimal activity at 20°C to 50°C which makes them unable to withstand such adverse conditions